# Maternal characteristics associated with referral to obstetrician-led care in low-risk pregnant women in the Netherlands: A retrospective cohort study

Susan Niessink-Beckers[1]*, Corine J. Verhoeven[1,2,3], Marleen J. Nahuis[4], Lisanne A. Horvat-Gitsels[5], Janneke T. Gitsels-van der Wal[1]

1 Amsterdam UMC, Vrije Universiteit Amsterdam, Midwifery Science, AVAG, Amsterdam Public Health Research Institute, Amsterdam, Netherlands, 2 Department of Obstetrics and Gynecology, Maxima Medical Center, Veldhoven, Netherlands, 3 Division of Midwifery, School of Health Sciences, University of Nottingham, Nottingham, United Kingdom, 4 Department of Obstetrics and Gynecology, Noordwest Hospital Group location Alkmaar, Alkmaar, Netherlands, 5 UCL Great Ormond Street Institute of Child Health, Faculty of Population Health Sciences, University College London, London, United Kingdom

* susanbeckers@gmail.com

**Data Availability Statement:** All relevant data are within the paper and its Supporting Information files.

## Abstract

### Background

In the Netherlands, maternity care is divided into midwife-led care (for low-risk women) and obstetrician-led care (for high-risk women). Referrals from midwife-led to obstetrician-led care have increased over the past decade. The majority of women are referred during their pregnancy or labour. Referrals are based on a continuous risk assessment of the health and characteristics of mother and child, yet referral for non-medical factors and characteristics remain unclear. This study investigated which maternal characteristics are associated with women's referral from midwife-led to obstetrician-led care.

### Materials and methods

A retrospective cohort study in one midwife-led care practice in the Netherlands included 1096 low-risk women during January 2015–17. The primary outcomes were referral from midwife-led to obstetrician-led care in (1) the antepartum period and (2) the intrapartum period. In total, 11 maternal characteristics were identified. Logistic regression models of referral in each period were fitted and stratified by parity.

### Results

In the antepartum period, referral among nulliparous women was associated with an older maternal age (aOR, 1.07; 95%CI, 1.05–1.09), being underweight (0.45; 0.31–0.64), over-weight (2.29; 1.91–2.74), or obese (2.65; 2.06–3.42), a preconception period >1 year (1.34; 1.07–1.66), medium education level (0.76; 0.58–1.00), deprivation (1.87; 1.54–2.26), and sexual abuse (1.44; 1.14–1.82). Among multiparous women, a referral was associated with being underweight (0.40; 0.26–0.60), obese (1.61; 1.30–1.98), a preconception period >1

**Funding:** The author(s) received no specific funding for this work.

**Competing interests:** The authors have declared that no competing interests exist.

year (1.71; 1.27–2.28), employment (1.38; 1.19–1.61), deprivation (1.23; 1.03–1.46), highest education level (0.63; 0.51–0.80), psychological problems (1.24; 1.06–1.44), and one or multiple consultations with an obstetrician (0.68; 0.58–0.80 and 0.64; 0.54–0.76, respectively). In the intrapartum period, referral among nulliparous women was associated with an older maternal age (1.02; 1.00–1.05), being underweight (1.67; 1.15–2.42), a preconception period >1 year (0.42; 0.31–0.57), medium or high level of education (2.09; 1.49–2.91 or 1.56; 1.10–2.22, respectively), sexual abuse (0.46; 0.33–0.63), and multiple consultations with an obstetrician (1.49; 1.15–1.94). Among multiparous women, referral was associated with an older maternal age (1.02; 1.00–1.04), being overweight (0.65; 0.51–0.83), a preconception period >1 year (0.33; 0.17–0.65), non-Dutch ethnicity (1.98; 1.61–2.45), smoking (0.75; 0.57–0.97), sexual abuse (1.49; 1.09–2.02), and one or multiple consultations with an obstetrician (1.34; 1.06–1.70 and 2.09; 1.63–2.69, respectively).

## Conclusions

This exploratory study showed that several non-medical maternal characteristics of low-risk pregnant women are associated with referral from midwife-led to obstetrician-led care, and how these differ by parity and partum period.

## Introduction

Multiple countries worldwide provide midwife-led care, e.g., Australia, Canada, New Zealand, the United Kingdom and the Netherlands [1]. Midwife-led care is a model where "the midwife is the lead professional in the planning, organisation and delivery of care given to a woman from initial booking to the postnatal period" [2]. Other models of care are obstetrician-led care, family doctor-led care or shared care. The Dutch maternity care system is divided into two echelons: primary and secondary care. In primary care, known as midwife-led care, midwives provide care for low-risk pregnant women during the antepartum, intrapartum and postpartum periods. Over 87% of all pregnant women in the Netherlands start their prenatal care in primary care [3]. Women will remain in primary care if they are healthy and no complications occur. In cases where pathology occurs in the antepartum, intrapartum, or postpartum period, women are referred to secondary care, also known as obstetrician-led care. In secondary care, the care will be provided by obstetricians or hospital-based midwives [4, 5]. In the Netherlands, the number of referrals in the intrapartum period has increased from 27% to 41% over the past 12 years [3, 6].

The List of Obstetric Indications (LOI) provides guidelines for determining whether a woman should receive midwife-led or obstetrician-led care, mainly based on medical and obstetric history [4, 7]. The main antepartum indications for a referral are gestational diabetes, pregnancy-induced hypertension, and previous caesarean section [3, 8]. The intrapartum indications include a request for medical pain relief such as epidural analgesia, the presence of meconium-stained amniotic fluid, and failure to progress in the first or second stage of labour [9–12]. In particular, the number of referrals for a request for medical pain relief has increased; in 2004, only 4% of women received epidural analgesia during labour compared to 21% in 2017 [3]. Dutch primary care midwives are not qualified to care for women who receive an epidural in the Netherlands. Therefore, they are referred to obstetrician-led care.

Numerous studies have determined the influence of risk factors such as body mass index (BMI) or maternal age on perinatal outcomes [13–17]. However, few studies have investigated the association between perinatal outcomes and a wide range of maternal characteristics that are readily available in clinical records, such as a history or presence of psychological problems or sexual abuse [18–25]. Moreover, the associations between specifically non-medical maternal factors and referral towards obstetric care are still unknown. Awareness of all maternal characteristics, including non-medical factors affecting the chance of a referral, will help healthcare professionals provide individualised preventive care [26, 27]. Before we can intervene on those factors, we need to know which non-medical factors increase the likelihood of referral to obstetrician-led care. Therefore, our study investigated which non-medical maternal characteristics are associated with women's referral from midwife-led to obstetrician-led care in the antepartum and intrapartum periods, as these may influence perinatal outcomes.

## Materials and methods

### Study design and participants

This retrospective cohort study took place in one large midwifery practice in an urban region near Amsterdam, the Netherlands. The study period was from January 2015 to January 2017 [8]. The study sample included women with a singleton pregnancy who received midwife-led care after the first trimester. Those who had a miscarriage were excluded and those who were referred to obstetrician-led care in the first trimester or received only postnatal care. Informed consent was obtained verbally and noted in their medical records in their presence [8]. The Medical Ethics Committee of the Amsterdam University Medical Centres (location VUmc) (FWA00017598) approved the study (ref. 2018.019).

### Measures

The two primary outcomes were defined as a referral from midwife-led to obstetrician-led care in the antepartum (yes/no) and, given no previous referral, in the intrapartum period (yes/no). Maternal non-medical characteristics—the independent variables of interest—were based on literature on the effect of these characteristics on morbidity and mortality of mother and child [13, 15–18, 21, 23–25, 28]. These characteristics were obtained from the women's medical records where they were noted by the midwife at the beginning of the antepartum period. Dichotomous variables included: preconception period (≤1 year/>1 year)—the period the woman was trying to conceive—, employment (yes/no), ethnicity (Dutch/non-Dutch)—based on the mothers country of birth—, lived in a deprived area (no/yes)—based on a zip code classified as a deprived area [29]—, smoking (no/yes), psychological problems (past and previous) (no/yes), and a history of sexual abuse (no/yes). Education level was categorised into low (primary education, pre-vocational secondary education or secondary vocational education), medium (senior general secondary education or pre-university education) and high (higher professional education or university education). BMI and number of consultations with an obstetrician were categorised due to non-normal distributions. Underweight was defined as having a BMI of <18.5, healthy weight was defined as a BMI of 18.5–24.9, overweight was defined as a BMI of 25.0–29.9, and obesity was defined as a BMI of ≥30.0 [30]. The number of consultations with an obstetrician—a standalone consult in obstetric care without women discontinuing midwife-led care—was defined as none, one, and ≥2. Maternal age was a continuous variable based on the date of birth.

Compared to multiparous women, nulliparous women are more likely to be referred to obstetrician-led care both during the antepartum and intrapartum periods [3, 9, 31]. Therefore, parity was identified as an effect modifier. The reasons for referral were grouped by the

antepartum and intrapartum periods and were based on regional protocols and the LOI. https://www.ncbi.nlm.nih.gov/pubmed/30153431

## Statistical analyses

The study population's baseline characteristics were summarised by means and standard deviations for normally distributed continuous variables, and frequencies and percentages for categorical variables, including dichotomous ones. Missing data patterns were explored by fitting logistic regression models to understand potential selection bias and dealt with by multiple imputations [32, 33]. Logistic regression models of referral in the antepartum and intrapartum periods were stratified by parity. The leanest models were obtained using backward elimination with a significance level set at α = 0.05. The logistic model assumptions of linearity, independence of errors, and multicollinearity were checked by looking at interactions between predictor and its log transformation, Durbin-Watson tests, and variance inflation factors, respectively. Model performance was assessed by specificity, sensitivity, and total accuracy [34–38]. The sensitivity analysis included repeating the model fittings on the subset with complete information (i.e., complete case analysis). All analyses were performed in SPSS version 24.

## Results

### Descriptive analysis

The study sample included 1096 participants with 520 (47%) nulliparous and 576 (53%) multiparous women. A total of 448 (41%) participants were referred to obstetrician-led care during the antepartum period, 39% of nulliparous and 42% of multiparous women. Among the 617 women who started labour in midwife-led care, 31 (5%) were moved to another practice or clinic in the country by the end of the antepartum period and 287 (47%) were referred to obstetrician-led care in the intrapartum period, 62% of nulliparous and 33% of multiparous women. The baseline characteristics by partum period and parity are presented in Table 1.

The most common reasons for referral in the antepartum period were gestational diabetes (13%) and pregnancy-induced hypertension (6%) (Fig 1A). In the intrapartum period, the most common reason for referral were a request for pain relief (13%) and a failure to progress in the first stage of labour (12%) (Fig 1B).

There was limited missing data (<1%), except for education (3%) and preconception period (7%), with overall 11% missing observations. The proportion of missing data did not differ by referral and parity in the antepartum period; aOR of 0.86 (95%CI, 0.46–1.57) for nulliparous women and 1.34 (0.81–2.21) for multiparous women. However, it was higher among multiparous women who were referred in the intrapartum period (2.52; 1.25–5.13). This means that in the complete-data sample of the intrapartum period, referred multiparous women were under-represented (S1 Table).

### Main analysis

**Antepartum.** The full and final (leanest) models of referral to obstetrician-led care in the antepartum period by parity are presented in Table 2A and their model performances in S2 Table; only the final models are described. Among nulliparous women, older women were more likely to be referred (aOR, 1.07; 95%CI, 1.05–1.09). Compared to women of healthy weight, those who were underweight were less likely to be referred (0.45; 0.31–0.64), whilst those who were overweight or obese were more likely to be referred (2.29; 1.91–2.74 or 2.65; 2.06–3.42, respectively). Women were also more likely to be referred when the preconception

**Table 1. Characteristics of the study population split by parity and referral period.**

*a: Characteristics of the study population concerning a referral in the antepartum period*

| | | Total | Parity | | | |
|---|---|---|---|---|---|---|
| | | | Nulliparous | | Multiparous | |
| | | Total | Not referred | Referred | Not referred | Referred |
| | | n = 1096 (100.0%) | n = 316 (28.8%) | n = 204 | n = 332 | n = 244 |
| | | | | (18.6%) | (30.3%) | (22.3%) |
| | | *mean (SD)* | *mean (SD)* | *mean (SD)* | *mean (SD)* | *mean (SD)* |
| Maternal age | | 29.2 (5.0) | 26.7 (4.6) | 28.3 (4.8) | 30.9 (4.4) | 31.1 (4.7) |
| | | *n (%)* | *n (% of no nulliparous)* | *n (% of yes nulliparous)* | *n (% of no multiparous)* | *n (% of yes multiparous)* |
| BMI* | | | | | | |
| | <18.5 | 61 (5.6) | 30 (9.6) | 7 (3.4) | 19 (5.8) | 5 (2.1) |
| | 18.5–24.9 | 611 (56.1) | 205 (65.3) | 100 (49.0) | 182 (55.2) | 124 (51.2) |
| | 25–29.9 | 282 (25.9) | 57 (18.2) | 65 (31.9) | 90 (27.3) | 70 (28.9) |
| | ≥30 | 136 (12.5) | 22 (7.0) | 32 (15.7) | 39 (11.8) | 43 (17.8) |
| Preconception period* | | | | | | |
| | ≤ 1 year | 920 (90.2) | 264 (88.9) | 159 (82.0) | 294 (95.5) | 203 (91.9) |
| | > 1 year | 100 (9.8) | 33 (11.1) | 35 (18.0) | 14 (4.5) | 18 (8.1) |
| Education level* | | | | | | |
| | low | 132 (12.5) | 31 (10.1) | 20 (10.1) | 44 (13.8) | 37 (15.9) |
| | medium | 501 (47.4) | 145 (47.1) | 81 (40.9) | 153 (47.8) | 122 (52.6) |
| | high | 425 (40.2) | 132 (42.9) | 97 (49.0) | 123 (38.4) | 73 (31.5) |
| Employment* | | | | | | |
| | yes | 734 (67.8) | 228 (73.1) | 153 (75.7) | 200 (60.6) | 153 (64.0) |
| | no | 349 (32.2) | 84 (26.9) | 49 (24.3) | 130 (39.4) | 86 (36.0) |
| Ethnicity* | | | | | | |
| | Dutch | 508 (46.4) | 162 (51.3) | 102 (50.7) | 141 (42.5) | 101 (42.6) |
| | non-Dutch | 586 (53.6) | 154 (48.7) | 99 (49.3) | 191 (57.5) | 136 (57.4) |
| Lived in deprived area* | | | | | | |
| | yes | 228 (20.8) | 49 (15.6) | 51 (25.0) | 69 (20.8) | 59 (24.3) |
| | no | 866 (79.2) | 266 (84.4) | 153 (75.0) | 263 (79.2) | 184 (75.7) |
| Smoking* | | | | | | |
| | yes | 253 (23.2) | 80 (25.4) | 54 (26.6) | 64 (19.3) | 55 (22.6) |
| | no | 839 (76.8) | 235 (74.6) | 149 (73.4) | 267 (80.7) | 188 (77.4) |
| Psychological problems* | | | | | | |
| | yes | 309 (28.3) | 83 (26.5) | 61 (29.9) | 90 (27.2) | 75 (30.9) |
| | no | 782 (71.7) | 230 (73.5) | 143 (70.1) | 241 (72.8) | 168 (69.1) |
| History of sexual abuse* | | | | | | |
| | yes | 135 (12.4) | 38 (12.1) | 29 (14.2) | 38 (11.5) | 30 (12.3) |
| | no | 957 (87.6) | 276 (87.9) | 175 (85.8) | 293 (88.5) | 213 (87.7) |
| Consultation obstetric care* | | | | | | |
| | none | 503 (46.1) | 142 (45.4) | 101 (49.8) | 136 (41.0) | 124 (50.8) |
| | 1 | 330 (30.2) | 100 (31.9) | 56 (27.6) | 108 (32.5) | 66 (27.0) |
| | > 1 | 259 (23.7) | 71 (22.7) | 46 (22.7) | 88 (26.5) | 54 (22.1) |

*b: Characteristics of the study population concerning a referral in the intrapartum period*

| | | Total | Parity | | | |
|---|---|---|---|---|---|---|
| | | | Nulliparous | | Multiparous | |
| | | Total | Not referred | Referred | Not referred | Referred |
| | | n = 617 (100.0%) | n = 112 (18.2%) | n = 179 (29.0%) | n = 218 (35.3%) | n = 108 (17.5%) |

*(Continued)*

**Table 1.** (Continued)

| | | mean (SD) | mean (SD) | mean (SD) | mean (SD) | mean (SD) |
|---|---|---|---|---|---|---|
| Maternal age | | 28.9 (4.8) | 26.6 (4.2) | 26.8 (4.6) | 30.8 (4.2) | 30.8 (4.5) |
| | | $n$ (%) | $n$ (% of no nulliparous) | $n$ (% of yes nulliparous) | $n$ (% of no multiparous) | $n$ (% of yes multiparous) |
| BMI* | | | | | | |
| | <18.5 | 45 (7.3) | 8 (7.2) | 18 (10.1) | 13 (6.0) | 6 (5.7) |
| | 18.5–24.9 | 372 (60.6) | 75 (67.6) | 117 (65.4) | 118 (54.1) | 62 (58.5) |
| | 25–29.9 | 141 (23.0) | 21 (18.9) | 31 (17.3) | 64 (29.4) | 25 (23.6) |
| | ≥30 | 56 (9.1) | 7 (6.3) | 13 (7.3) | 23 (10.6) | 13 (12.3) |
| Preconception period* | | | | | | |
| | ≤ 1 year | 533 (92.5) | 89 (83.2) | 153 (92.2) | 193 (95.1) | 98 (98.0) |
| | > 1 year | 43 (7.5) | 18 (16.8) | 13 (7.8) | 10 (4.9) | 2 (2.0) |
| Education level* | | | | | | |
| | low | 72 (12.0) | 17 (15.2) | 13 (7.5) | 28 (13.0) | 14 (14.1) |
| | medium | 284 (47.4) | 47 (42.0) | 87 (50.3) | 96 (44.7) | 54 (54.5) |
| | high | 243 (40.6) | 48 (42.9) | 73 (42.2) | 91 (42.3) | 31 (31.3) |
| Employment* | | | | | | |
| | yes | 410 (67.0) | 84 (75.0) | 130 (73.9) | 135 (62.2) | 61 (57.0) |
| | no | 202 (33.0) | 28 (25.0) | 46 (26.1) | 82 (37.8) | 46 (43.0) |
| Ethnicity* | | | | | | |
| | Dutch | 288 (46.7) | 60 (55.6) | 90 (50.3) | 124 (57.7) | 34 (31.5) |
| | non-Dutch | 329 (53.3) | 48 (44.4) | 89 (49.7) | 91 (42.3) | 74 (68.5) |
| Lived in deprived area* | | | | | | |
| | yes | 116 (18.8) | 17 (15.2) | 31 (17.3) | 41 (18.8) | 27 (25.0) |
| | no | 501 (81.2) | 95 (84.8) | 148 (82.7) | 177 (81.2) | 81 (75.0) |
| Smoking* | | | | | | |
| | yes | 136 (22.1) | 27 (24.1) | 46 (25.7) | 43 (19.8) | 20 (18.5) |
| | no | 480 (77.9) | 85 (75.9) | 133 (74.3) | 174 (80.2) | 88 (81.5) |
| Psychological problems* | | | | | | |
| | yes | 165 (26.9) | 31 (27.9) | 46 (25.8) | 56 (25.7) | 32 (29.9) |
| | no | 449 (73.1) | 80 (72.1) | 132 (74.2) | 162 (74.3) | 75 (70.1) |
| History of sexual abuse* | | | | | | |
| | yes | 68 (11.1) | 17 (15.3) | 14 (7.8) | 22 (10.1) | 15 (14.0) |
| | no | 547 (88.9) | 94 (84.7) | 165 (92.2) | 196 (89.9) | 92 (86.0) |
| Consultation obstetric care* | | | | | | |
| | none | 256 (41.7) | 50 (45.0) | 73 (41.2) | 99 (45.4) | 34 (31.5) |
| | 1 | 200 (32.6) | 36 (32.4) | 58 (32.8) | 71 (32.6) | 35 (32.4) |
| | >1 | 158 (25.7) | 25 (22.5) | 46 (26.0) | 48 (22.0) | 39 (36.1) |

*Sample size varies due to missing data; valid percentages are shown.

BMI: body mass index

period was longer than a year (1.34; 1.07–1.66), they lived in a deprived area (1.87; 1.54–2.26), and had a history of sexual abuse (1.44; 1.14–1.82).

Among multiparous women, compared to those with a healthy weight, underweight women were less likely to receive a referral (0.40; 0.26–0.60) than obese women (1.61; 1.30–1.98). Women were also more likely to be referred if their preconception period was longer than one year (1.71; 1.27–2.28), they had a history of psychological problems (1.24; 1.06–1.44), they worked during pregnancy (1.38; 1.19–1.61), and lived in a deprived area (1.23; 1.03–1.46).

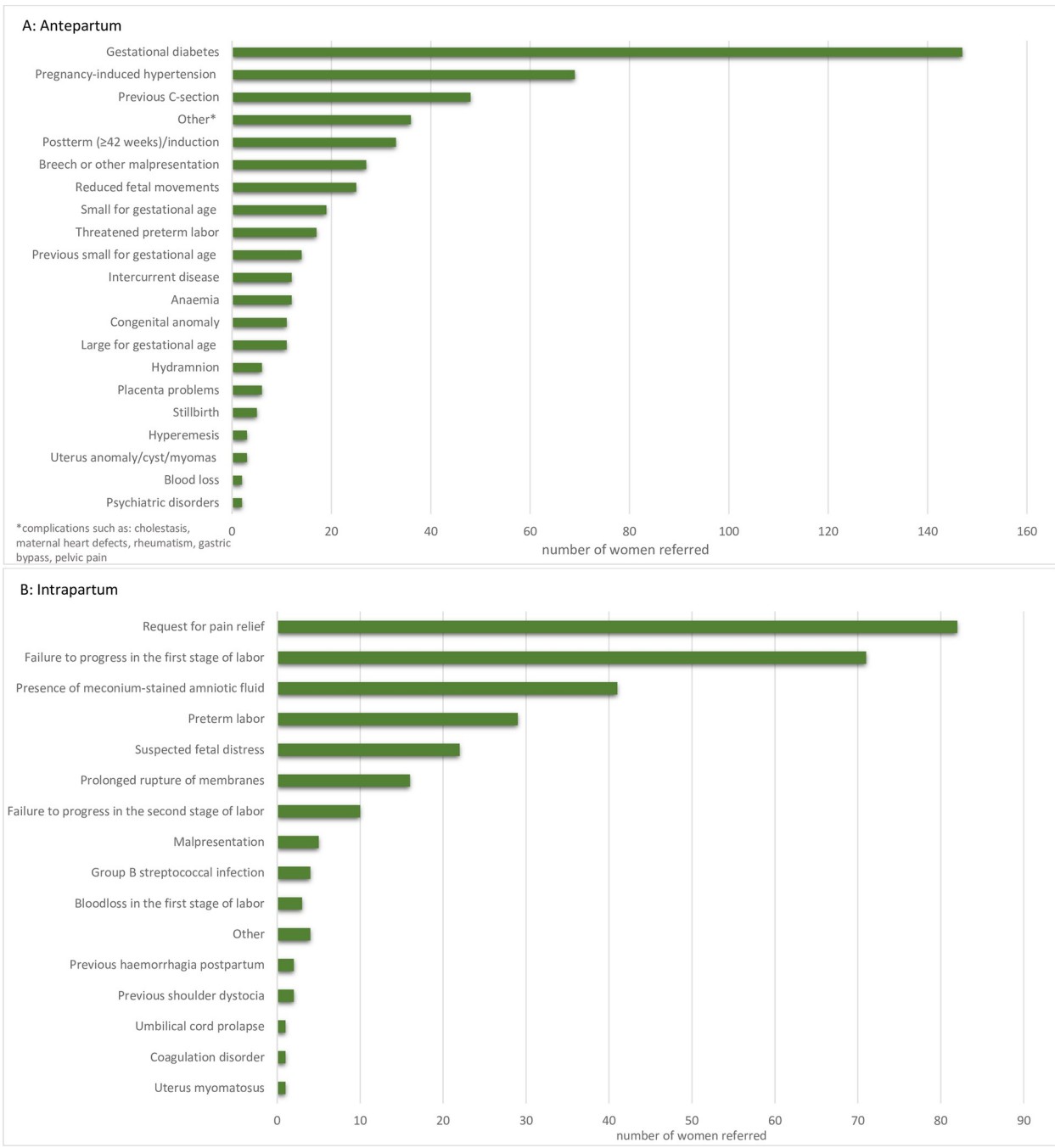

**Fig 1. Reasons for referral antepartum & intrapartum.** (A) Reasons for referral in the antepartum period. (B) Reasons for referral in the intrapartum period.

In contrast, women were less likely to be referred when they had a high as opposed to a low education level (0.63; 0.51–0.80), and had one or more consultations in obstetric care (0.68; 0.58–0.80 or 0.64; 0.54–0.76, respectively).

**Intrapartum.** The full and final (leanest) models of referral to obstetrician-led care in the intrapartum period by parity are presented in Table 2B and their model performances in S2 Table; only the final models are described. Among nulliparous women, older women had

**Table 2. Association with referral split by parity and period.**

*a*: *Association with referral antepartum (n = 1096)*

| Variable | Nulliparous | | Multiparous | |
|---|---|---|---|---|
| | *Full model* | *Final model* | *Full model* | *Final model* |
| | OR (95%CI) | OR (95%CI) | OR (95%CI) | OR (95%CI) |
| Maternal age (years) | 1.07 (1.05–1.09) | **1.07 (1.05–1.09)** * | 1.00 (0.99–1.02) | |
| BMI (kg/m$^2$) | | | | |
| <18.5 | 0.45 (0.31–0.64) | **0.45 (0.31–0.64)** * | 0.38 (0.25–0.58) | **0.40 (0.26–0.60)** * |
| 18.5–24.9 | 1.00 | 1.00 | 1.00 | 1.00 |
| 25.0–29.9 | 2.27 (1.89–2.72) | **2.29 (1.91–2.74)** * | 1.08 (0.92–1.27) | 1.08 (0.92–1.27) |
| ≥ 30.0 | 2.64 (2.05–3.41) | **2.65 (2.06–3.42)** * | 1.60 (1.30–1.97) | **1.61 (1.30–1.98)** * |
| Preconception period (≤ 1 year/> 1 year) | 1.35 (1.08–1.69) | **1.34 (1.07–1.66)** * | 1.70 (1.27–2.28) | **1.71 (1.27–2.28)** * |
| Education level | | | | |
| Low | 1.00 | 1.00 | 1.00 | 1.00 |
| Medium | 0.74 (0.57–0.98) | **0.76 (0.58–1.00)** * | 0.88 (0.71–1.08) | 0.86 (0.70–1.06) |
| High | 0.93 (0.70–1.24) | 0.94 (0.71–1.24) | 0.65 (0.52–0.82) | **0.63 (0.51–0.80)** * |
| Employment (no/yes) | 1.09 (0.90–1.33) | | 1.43 (1.21–1.68) | **1.38 (1.19–1.61)** * |
| Ethnicity (Dutch/non-Dutch) | 0.98 (0.83–1.17) | | 1.08 (0.92–1.27) | |
| Lived in a deprived area (no/yes) | 1.89 (1.54–2.32) | **1.87 (1.54–2.26)** * | 1.20 (1.01–1.44) | **1.23 (1.03–1.46)** * |
| Smoking (no/yes) | 1.10 (0.92–1.33) | | 1.12 (0.94–1.34) | |
| Psychological problems (no/yes) | 0.93 (0.77–1.12) | | 1.20 (1.01–1.42) | **1.24 (1.06–1.44)** * |
| History of sexual abuse (no/yes) | 1.48 (1.15–1.90) | **1.44 (1.14–1.82)** * | 1.14 (0.90–1.44) | |
| Consultation obstetric care | | | | |
| None | 1.00 | | 1.00 | 1.00 |
| 1 | 0.88 (0.73–1.05) | | 0.68 (0.58–0.81) | **0.68 (0.58–0.80)** * |
| > 1 | 0.93 (0.77–1.14) | | 0.64 (0.53–0.76) | **0.64 (0.54–0.76)** * |

*b*: *Association with referral intrapartum (n = 617)*

| Variable | Nulliparous | | Multiparous | |
|---|---|---|---|---|
| | *Full model* | *Final model* | *Full model* | *Final model* |
| | OR (95%CI) | OR (95%CI) | OR (95%CI) | OR (95%CI) |
| Maternal age (years) | 1.03 (1.01–1.04) | **1.02 (1.00–1.05)** * | 1.02 (1.00–1.05) | **1.02 (1.00–1.04)** * |
| BMI (kg/m$^2$) | | | | |
| <18.5 | 1.71 (1.18–2.48) | **1.67 (1.15–2.42)** * | 0.86 (0.56–1.33) | 0.84 (0.55–1.30) |
| 18.5–24.9 | 1.00 | 1.00 | 1.00 | 1.00 |
| 25.0–29.9 | 1.00 (0.76–1.31) | 0.98 (0.75–1.28) | 0.65 (0.51–0.83) | **0.65 (0.51–0.83)** * |
| ≥ 30.0 | 1.15 (0.76–1.74) | 1.14 (0.75–1.73) | 0.84 (0.60–1.16) | 0.83 (0.60–1.15) |
| Preconception period (≤ 1 year/> 1 year) | 0.42 (0.30–0.57) | **0.42 (0.31–0.57)** * | 0.35 (0.18–0.68) | **0.33 (0.17–0.65)** * |
| Education level | | | | |
| Low | 1.00 | 1.00 | 1.00 | |
| Medium | 2.20 (1.56–3.11) | **2.09 (1.49–2.91)** * | 1.26 (0.92–1.74) | |
| High | 1.74 (1.20–2.52) | **1.56 (1.10–2.22)** * | 0.79 (0.56–1.11) | |
| Employment (no/yes) | 0.83 (0.64–1.08) | | 1.13 (0.90–1.43) | |
| Ethnicity (Dutch/non-Dutch) | 1.05 (0.84–1.32) | | 1.98 (1.58–2.49) | **1.98 (1.61–2.45)** * |
| Lived in a deprived area (no/yes) | 1.05 (0.79–1.39) | | 1.21 (0.93–1.56) | |
| Smoking (no/yes) | 1.16 (0.90–1.48) | | 0.73 (0.56–0.95) | **0.75 (0.57–0.97)** * |
| Psychological problems (no/yes) | 0.96 (0.75–1.23) | | 1.07 (0.85–1.36) | |
| History of sexual abuse (no/yes) | 0.46 (0.32–0.65) | **0.46 (0.33–0.63)** * | 1.50 (1.08–2.09) | **1.49 (1.09–2.02)** * |
| Consultation obstetric care | | | | |
| None | 1.00 | 1.00 | 1.00 | 1.00 |

*(Continued)*

**Table 2.** (Continued）

| | | | | |
|---|---|---|---|---|
| 1 | 1.09 (0.86–1.38) | 1.10 (0.87–1.39) | 1.32 (1.04–1.69) | **1.34 (1.06–1.70)*** |
| > 1 | 1.44 (1.10–1.88) | **1.49 (1.15–1.94)*** | 2.08 (1.62–2.66) | **2.09 (1.63–2.69)*** |

* Statically significant (p<0.05)

BMI: body mass index

increased odds of being referred (1.02; 1.00–1.05). Compared to women of healthy weight, those who were underweight were more likely to be referred (1.67; 1.15–2.42) as well as women with multiple consultations during the pregnancy (1.49; 1.15–1.94). Women who had a higher education level had increased odds of a referral compared to the lowest level: medium education level (2.09; 1.49–2.91) and high education level (1.56; 1.10–2.22). Women were less likely to be referred when their preconception period was more than one year (0.42; 0.31–0.57) or they had a history of sexual abuse (0.46; 0.33–0.63).

Among multiparous women, older women had increased odds of being referred (1.02; 1.00–1.04). Women were more likely to be referred when they were from a non-Dutch ethnic group (1.98; 1.61–2.45), had a history of sexual abuse (1.49; 1.09–2.02), or had one or more consultations in obstetrician-led care (1.34; 1.06–1.70 or 2.09; 1.63–2.69, respectively). The odds of being referred were lower in those who were overweight compared to those being of a healthy weight (0.65; 0.51–0.83), had a preconception period of more than one year (0.33; 0.17–0.65), or were smokers (0.75; 0.57–0.97).

## Discussion

Our study investigated the possible associations between multiple maternal characteristics and referral from midwife-led to obstetrician-led care during the antepartum and intrapartum periods. In addition to the more commonly researched maternal characteristics such as BMI and age, we showed that non-medical characteristics such as employment, education level, history of sexual abuse and consultations in obstetrician-led care are associated with referral, and differ by parity and partum period. Overall, maternal characteristics associated with referral during the antepartum period were age, BMI, preconception period, education level, employment, living in a deprived area, psychological problems, history of sexual abuse, and consultations in obstetric care. In the intrapartum period, maternal characteristics associated with referral were age, BMI, preconception period, education level, ethnicity, smoking, history of sexual abuse, and consultations in obstetrician-led care.

In agreement with other studies, we found that age, BMI, ethnicity, smoking and living in a deprived area are associated with referral [18, 23, 25, 39–41]. A meta-analysis found that women with a preconception period of more than one year had an increased risk for preterm birth, low birth weight and small-for-gestational-age [21], which are all reasons for referral and thus agree with our findings in the antepartum period. However, in the intrapartum period, the effect was the opposite, which might be explained by the fact that over 50% of these women were already referred to obstetrician-led care in the antepartum period.

The number of consultations in obstetrician-led care (without transfer of care) had a mixed effect on referral to obstetrician-led care; this increased the likelihood of referral for both nulliparous and multiparous women in the intrapartum period, whereas in the antepartum period it had no significant effect for nulliparous women and a decreased effect for multiparous women. This has not been studied before, but we hypothesise that it comes down to the different reasons for consultations in obstetrician-led care. For instance, multiparous women with a

history of postpartum haemorrhage or manual removal of the placenta need a consultation during their current pregnancy. However, they were not at risk for medical complications in the antepartum period.

Multiparous but not nulliparous women who worked during pregnancy had higher odds of being referred to obstetrician-led care in the antepartum period. Although we cannot explain the different effects by parity, a meta-analysis showed that physically demanding work is significantly associated with hypertension or preeclampsia and preterm birth, which are reasons for referral [22].

Nulliparous women with a medium or high education level compared to women with a low education level had a higher chance of being referred to obstetric care in the intrapartum period. This is contrary to other studies that demonstrate a lower education level is associated with more medical complications during the intrapartum period [19, 23]. The relatively small sample size could be a reason for this contradicting result, as only 27 women had a low education level.

A history of sexual abuse had a mixed effect on referral to obstetrician-led care; during the antepartum period, it was associated with higher odds for nulliparous women and no association for multiparous women, whilst during the intrapartum period it was associated with lower odds for nulliparous women and higher odds for multiparous women. This is not consistent with the literature, which shows that a history of sexual abuse is associated with more psychological problems and adverse perinatal outcomes [20, 42].

Overall, for the antepartum and intrapartum periods, the associations between maternal characteristics and a referral differed. Most of the characteristics in the antepartum and intrapartum periods were comparable, except for a BMI$\geq$ 30; in the antepartum period, 13% of the women had a BMI $\geq$ 30 compared to 9% of the women in the intrapartum period. Since a BMI $\geq$ 30 is associated with more complications during pregnancy, these women might already have been referred to obstetrician-led care, which could explain the difference in the associations [17].

## Strength and limitations

The strength of our study is the availability of a large number of maternal characteristics, including non-medical ones such as psychosocial and lifestyle factors. Data were collected directly from the midwife practice's medical records rather than from routinely registered data, resulting in more reliable data.

Our low-risk study population included only one midwifery practice. Nationally, the rate of referrals varies by practice and location. A nationwide retrospective cohort study reported that intrapartum referrals of nulliparous or multiparous women range between 55–68% and 20–32%, respectively (in our study 62% and 33%, respectively). The care providers' assessment of risk and uncertainty, as well as regional guidelines, may impact the referral rate [43]. Therefore, we cannot generalise our results to all low-risk women in the Netherlands. Our study had a higher proportion of non-Dutch women and smokers compared to the Dutch maternity population, and this could lead to more referrals to obstetrician-led care [44–47]. During our study period, all women with gestational diabetes were referred to obstetrician-led care. The current policy in this region is that women who only need a diet to stabilise their glucose values are not referred to obstetrician-led care. This policy is no longer in use in this region. Finally, our study had limited missing data, which were appropriately addressed by multiple imputations [32, 33].

## Practical implications and future research

Our study showed that multiple maternal characteristics are associated with a referral to obstetrician-led care in this particular midwifery practice. In particular, there are a number of non-

medical characteristics associated with a referral that can affect the course of pregnancy and birth, but these are currently not considered in maternity care [26, 27]. Non-medical issues could be addressed at an earlier stage, preferably during preconception or at the beginning of pregnancy. For instance, addressing the lack of social support is important during pregnancy, as this is associated with adverse perinatal outcomes [48, 49]. Therefore, we would advocate more facilities for midwives, as they often provide close care to women and know their social environment. This would allow midwives to treat women with non-medical factors more intensely.

Care models such as midwife-led care, which emphasise continuity of care, are in themselves important for the well-being of the mother and child [1]; and other midwife-led care models such as CenteringPregnancy (group sessions), facilitate social support and enable women to improve their self-confidence [50]. The women in this study had access to other caregivers. Continuity of care might be beneficial for non-medical maternal characteristics, particularly since case-load midwifery care is associated with a lower referral rate [51]. Support from other caregivers is also accessible in the midwife-led care model, whereby the midwife can support women with the help of a psychologist, dietitian, or welfare worker. As preventive care focused on non-medical issues may benefit medical care, we advocate further research into the relationship between non-medical maternal characteristics and referral, as well as how midwives can improve midwife-led care for women with psychosocial or lifestyle issues (non-medical characteristics).

## Conclusion

To our knowledge, this is the first study that illustrates a large number of non-medical maternal characteristics of low-risk pregnant women that are associated with a referral from midwife-led to obstetrician-led care in the antepartum and intrapartum periods, both for nulliparous and multiparous women. In particular, certain characteristics such as living in a deprived area, unemployment and a history of sexual abuse might benefit from other care models or interventions. These include case-load care or resilience-enhancing interventions such as CenteringPregnancy, as well as the supportive care of a welfare worker or psychologist. We advocate further research to increase awareness of the influence non-medical characteristics have on referral, as well as research about interventions that could improve modifiable maternal characteristics in the preconception and/or antepartum period.

## Supporting information

**S1 Table. Association between missing data in characteristics and referral.**
(DOCX)

**S2 Table. Association with referral split by parity and period (original data)–Tables a and b–this file depicts the original unimputed data.**
(DOCX)

**S1 File. STROBE statement—checklist of items that should be included in reports of observational studies.**
(DOCX)

## Acknowledgments

The authors would like to thank Dr Vicky S Adams for proofreading and the two anonymous reviewers for their insightful suggestions and careful reading of the manuscript.

## Ethics approval and consent to participate

Informed consent was obtained verbally and noted in women's medical records under women's supervision [8]. All methods were performed in accordance with the relevant guidelines and regulations (Declaration of Helsinki).

## Author Contributions

**Conceptualization:** Janneke T. Gitsels-van der Wal.

**Data curation:** Janneke T. Gitsels-van der Wal.

**Methodology:** Lisanne A. Horvat-Gitsels.

**Supervision:** Corine J. Verhoeven, Janneke T. Gitsels-van der Wal.

**Writing – original draft:** Susan Niessink-Beckers.

**Writing – review & editing:** Corine J. Verhoeven, Marleen J. Nahuis, Lisanne A. Horvat-Gitsels, Janneke T. Gitsels-van der Wal.

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
