## [Decision Letter · Decision Letter 0]

26 Apr 2022

PONE-D-21-28982Maternal characteristics as indicators of referral to obstetric-led care in low-risk pregnant women in the Netherlands a retrospective cohort studyPLOS ONE

Dear Dr. Niessink-Beckers,

Thank you for submitting your manuscript to PLOS ONE. After careful consideration, we feel that it has merit but does not fully meet PLOS ONE’s publication criteria as it currently stands. Therefore, we invite you to submit a revised version of the manuscript that addresses the points raised during the review process.

 As you were informed, there was a change of academic editor prior to evaluation. After reassignment, two reviewers experts in the field have provided their detailed comments that suggest the need of changes, particularly in terms of interpretation of results. Note also that, in order to ensure completeness of reporting, PLOS ONE recommends the use of checklists such as the Strobe checklist for cohort studies, https://www.strobe-statement.org/download/strobe-checklist-cohort-studies-pdf  and the RECORD checklist for REporting of studies Conducted using Observational Routinely-collected Data, https://www.record-statement.org/ Please submit your revised manuscript by Jun 10 2022 11:59PM. If you will need more time than this to complete your revisions, please reply to this message or contact the journal office at plosone@plos.org. Please include the following items when submitting your revised manuscript:A rebuttal letter that responds to each point raised by the academic editor and reviewer(s). You should upload this letter as a separate file labeled 'Response to Reviewers'.A marked-up copy of your manuscript that highlights changes made to the original version. You should upload this as a separate file labeled 'Revised Manuscript with Track Changes'.An unmarked version of your revised paper without tracked changes. You should upload this as a separate file labeled 'Manuscript'.

We look forward to receiving your revised manuscript.

Kind regards,

José Antonio Ortega, Ph.D.

Academic Editor

PLOS ONE

Journal Requirements:

Reviewers' comments:

Reviewer's Responses to Questions

**Comments to the Author**

1. Is the manuscript technically sound, and do the data support the conclusions?

Reviewer #1: Partly

Reviewer #2: Partly

2. Has the statistical analysis been performed appropriately and rigorously? 

Reviewer #1: Yes

Reviewer #2: I Don't Know

3. Have the authors made all data underlying the findings in their manuscript fully available?

Reviewer #1: Yes

Reviewer #2: Yes

4. Is the manuscript presented in an intelligible fashion and written in standard English?

Reviewer #1: Yes

Reviewer #2: No

5. Review Comments to the Author

Reviewer #1: Overall

This is a retrospective cohort study in one midwifery practice in the Netherlands, which investigates the association of various maternal characteristics (including psychosocial and lifestyle characteristics) with the chance on a referral from primary midwifery to secondary obstetric care. The strength of the study is the richness of the dataset, in comparison with the national perinatal database in the Netherlands. The associations found are informative for professionals as well for pregnant women and underpins that certain psycho-social and lifestyle factors are associated with referrals to obstetric care.

The reason for a higher chance on referrals is of course, that such problems affect her medical risks in pregnancy. A referral indicates that such a medical problem has indeed arisen. After the referral, her medical/obstetrical problem will be addressed, but this will probably not resolve or reduce the psycho-social or lifestyle issue.

Major issues

First, I express my concern about the general purpose of this article. By using the word ‘indicator’, and by presenting sensitivity and specificity, the authors suggest that they want to predict referrals based on medical and non-medical characteristics, in order to support midwives in decisions on referrals (so called ‘risk assessment’). However, I doubt whether better prediction of referrals benefits the care for women. In the Dutch context, a referral to secondary care is needed if a problem arises that need attention of an obstetric professional. Even when a woman is at elevated risk for obstetric problems based on her psycho-social or life style characteristics, an actual referral is needed only when this obstetric problem becomes evident. The prediction can be false, (total accuracy is lower than 70% in table 3), even if sensitivity and specificity are improved. Predicting a referral and acting on it by referring a woman based on the prediction creates a self-fulfilling prophecy.

In my opinion, the strength of this article is not that it may contribute to better prediction or risk assessment, but that it shows once more that certain non-medical issues have impact on the course of pregnancy and birth. These issues need the attention of the midwife, mainly because these issues are in itself important for the well-being of the mother and her family. The midwife can support women with these issues, or she eventually can suggest the help of a psychologist, dietitian or welfare worker. In my opinion, this should be discussed more extensively in the section on implications for practice (Line 264), instead of the current focus on improving risk assessment. For instance, it would be interesting to know how the midwifery practice in this study deals with such issues or what they think is necessary to improve midwifery care for women with psycho-social or life-style issues.

Selective group in intrapartum care

You present the association of characteristics with antepartum referrals and with intrapartum referral. Every women that experienced an antepartum referral, is no longer in midwifery care and therefor is no longer ‘at risk’ for an intrapartum referral. As the antepartum referrals are associated with certain characteristics, the intrapartum group (table 1b) has a different composition. For instance, in the antepartum group 12,5% of women have a BMI > 30 , and in the intrapartum group this is 9.1%. This selection process may partly explain some of your results in the intrapartum analyses, as you state in Line 219 for the characteristic ‘preconception period > 1yr’. But also education levels (Line 231) and sexual violence (Line 236) show unexpected results. Perhaps because the women with the most serious impact on their health were already referred in the antepartum period? Please elaborate on this in the discussion.

L224 You elaborate on the mixed / reverse relationship in the antepartum model for women with one or more consultations. Multiparous women with one or more consultations with obstetricians had a lower chance on an antepartum referral. That is contra-intuitive because consultations indicate that some kind of medical problem was present. Can this be explained by the way consultation are registered? Consultations that end up in a referral are probably not counted as a consultation but as a referral. If that is the case, please explain how this result should be interpreted in the antepartum analysis. In the intrapartum analyses the use of this variable make sense: it indicates that there was some kind of (medical) problem, but not severe enough to arrange a referral to obstetric care.

Minor issues

L 124 Maternal age is a continuous variable. Did you consider categorising this variable? There may well be a non-linear or (u-shaped) relation in your data, with higher complications and referrals for young (teens) and older women (> 35 or 40 yrs)

L 144 Among the 617 women who started labour in midwifery led care, 287 (47%) were referred to obstetric-led care, 62% of nulliparous and 33% of multiparous women. In the introduction, you describe percentages of 21%, probably with another denominator (all women, instead of women who started labour in midwifery led care). This is somewhat confusing.

Other:

Table 1: consider to organise the columns differently and group them by parity. This makes it easier to compare women of the same parity without referral with women with referral.

Line 239 “We cannot explain the results as research has consistently shown that a history of sexual violence is associated with perinatal outcomes (41).” Reference 41 (van der Hulst ea, 2006 ) is perhaps not the best study to refer to, as it is not an example of a study that show this association. On the contrary, this study shows no important differences in referrals and birth-interventions for women with a history of sexual violence who were in primary midwifery care in the Netherlands.

Reviewer #2: Please see my review in the attachement for more details. The paper could benefint from both textual and English laguage editing. My knowledge on statistical analysis is not brilliant but I do miss a statistical comparison of groups (see letter).

6. PLOS authors have the option to publish the peer review history of their article (what does this mean?). If published, this will include your full peer review and any attached files.

Reviewer #1: No

Reviewer #2: No

---

## [Author Response · Author response to Decision Letter 0]

21 Jun 2022

Please refer to the 'response to reviewers' document as part of the attached files and as requested.

Kind regards,

Susan Niessink-Beckers

---

## [Decision Letter · Decision Letter 1]

29 Jul 2022

PONE-D-21-28982R1Maternal characteristics associated with referral to obstetrician-led care in low-risk pregnant women in the Netherlands: a retrospective cohort studyPLOS ONE

Dear Dr. Niessink-Beckers,

Thank you for submitting your manuscript to PLOS ONE. After careful consideration, we feel that it has merit but does not fully meet PLOS ONE’s publication criteria as it currently stands. Therefore, we invite you to submit a revised version of the manuscript that addresses the points raised during the review process. The same two experts from the first round accepted to review the revised version. Reviewer 1 notes specific minor points to be enhanced. Reviewer 2 acknowledges that the suggested revisions have been implemented, but suggests rejection. This is mostly based on perceived contribution to the literature which is not one of PLOS ONE criteria. The manuscript could benefit, anyway, from a clearer explanation of its purpose and contribution. Note that reviewer 2 also provides, in a separate file, detailed feedback on specific points to improve in the manuscript.

We look forward to receiving your revised manuscript.

Kind regards,

José Antonio Ortega, Ph.D.

Academic Editor

PLOS ONE

Journal Requirements:

Reviewers' comments:

Reviewer's Responses to Questions

**Comments to the Author**

1. If the authors have adequately addressed your comments raised in a previous round of review and you feel that this manuscript is now acceptable for publication, you may indicate that here to bypass the “Comments to the Author” section, enter your conflict of interest statement in the “Confidential to Editor” section, and submit your "Accept" recommendation.

Reviewer #1: All comments have been addressed

Reviewer #2: All comments have been addressed

2. Is the manuscript technically sound, and do the data support the conclusions?

Reviewer #1: Yes

Reviewer #2: No

3. Has the statistical analysis been performed appropriately and rigorously? 

Reviewer #1: Yes

Reviewer #2: I Don't Know

4. Have the authors made all data underlying the findings in their manuscript fully available?

Reviewer #1: Yes

Reviewer #2: Yes

5. Is the manuscript presented in an intelligible fashion and written in standard English?

Reviewer #1: Yes

Reviewer #2: No

6. Review Comments to the Author

Reviewer #1: All comments are addressed in this new version. It is now an interesting paper, creating awareness for non-medical issues that need attention in perinatal care.

I only have some minor remarks:

Line 80:

insert: "Primary care midwives" instead of) 'Midwives' are not qualified taking care of women who receive an epidural in the Netherlands, therefor they are referred to obstetrician-led care (instead of "the obstetrician').

Line 87

To be able to intervene on those factors, we need to know which kind of non-medical factors are increasing (Insert: "the chance of") referral to obstetrician-led care

Line 172; table 2a/2b

Please replace ‘Indicators for referral’ with "association with referral "

Line 196

Remove the ; in this line after "or they had ; "

Lin2 271 “Lack of social support is associated with adverse perinatal outcomes and should be considered as an opportunity to intervene on psychosocial and lifestyle maternal characteristics [48, 49].”

This is a bit confusing. I suggest splitting this sentence and changing it a little into:

Lack of social support is associated with adverse perinatal outcomes. For these women, pregnancy should be considered …etc.

Line 287 “care systems” seems not the appropriate concept, I suggest “care models” .

Line 289: Do you really mean “all characteristics” or do you want to focus on non-medical modifiable characteristics (since medical characteristics already have the attention of maternity care professionals) ?

Reviewer #2: You have addressed all the comments diligently, thank you and I appreciate the change of tone in the discussion section. towards how we can best support women during pregnancy and labour.

I am still not clear what your aim was with this study and I am in doubt whether it will add anything substantial to the current discussion about referral rates, and my impression is that you fail to rise above the collected data.

7. PLOS authors have the option to publish the peer review history of their article (what does this mean?). If published, this will include your full peer review and any attached files.

Reviewer #1: No

Reviewer #2: No

---

## [Author Response · Author response to Decision Letter 1]

25 Aug 2022

Dear Editor and reviewers, 

Thank you for the expert review of our revised manuscript titled ‘Maternal characteristics associated with referral to obstetrician-led care in low-risk pregnant women in the Netherlands: a retrospective cohort study’.

We want to thank you and the reviewers for the second round of comments. We have used the reviewers' comments to improve our paper further, and we are pleased to send you our revised manuscript. The reviewers' comments and our responses are presented in the table below. The actual changes in the paper are presented in track changes. Furthermore, as suggested by Reviewer 2, a native speaker revised the English language.

Sincerely, on behalf of all authors,

Susan Niessink-Beckers

---

## [Decision Letter · Decision Letter 2]

16 Jan 2023

PONE-D-21-28982R2Maternal characteristics associated with referral to obstetrician-led care in low-risk pregnant women in the Netherlands: a retrospective cohort studyPLOS ONE

Dear Dr. Niessink-Beckers,

Thank you for submitting your manuscript to PLOS ONE. After careful consideration, we feel that it has merit but does not fully meet PLOS ONE’s publication criteria as it currently stands. Therefore, we invite you to submit a revised version of the manuscript that addresses the points raised during the review process.

Thank you very much for attending to all previous reviewer queries. Both reviewers are happy with the revisions, but have offered some additional very minor suggestions on the latest version, which you can find below. We wanted to give you a chance to address these comments, as we feel that doing so would further strengthen the manuscript.

We look forward to receiving your revised manuscript.

Kind regards,

Hanna Landenmark

Staff Editor

PLOS ONE

Journal Requirements:

Reviewers' comments:

Reviewer's Responses to Questions

**Comments to the Author**

1. If the authors have adequately addressed your comments raised in a previous round of review and you feel that this manuscript is now acceptable for publication, you may indicate that here to bypass the “Comments to the Author” section, enter your conflict of interest statement in the “Confidential to Editor” section, and submit your "Accept" recommendation.

Reviewer #1: All comments have been addressed

Reviewer #2: All comments have been addressed

2. Is the manuscript technically sound, and do the data support the conclusions?

Reviewer #1: Yes

Reviewer #2: Yes

3. Has the statistical analysis been performed appropriately and rigorously? 

Reviewer #1: Yes

Reviewer #2: I Don't Know

4. Have the authors made all data underlying the findings in their manuscript fully available?

Reviewer #1: Yes

Reviewer #2: Yes

5. Is the manuscript presented in an intelligible fashion and written in standard English?

Reviewer #1: Yes

Reviewer #2: Yes

6. Review Comments to the Author

Reviewer #1: I thank the authors for addressing all comments, the article is almost ready for publication. I have only one small remark: would you please check the subtitles in table 1a and 1b, where the % in the columns is explained?

Reviewer #2: Thank you for sending me this much improved paper on the study of maternal characteristics in primary midwifery care. I would really like to compliment the authors. The text reads much better and I am especially pleased with the improvements in the discussion and conclusion.

I still have a few minor remarks that I trust won’t take much effort to adjust:

Page 4, last Alinea of introduction: Your focus and your reasons for wanting to carry out this study are now much better explained. However I still have some trouble with the first 2 sentences of this Alinea, they seem to be contradictory. Are lifestyle factors (mentioned in the 2nd sentence) not factors such as BMI and maternal age? And there have been quite few studies so far that have shown an association between stress (psychosocial) and perinatal outcomes.

Moreover l87: …and maternal characteristics on non-medical factors > I think it should be: … and non-medical maternal characteristics?

P5 L101: I think there are words missing after “and”: Those who had a miscarriage were excluded and were referred to obstetrician-led care in the first trimester etc. This should read: and those who were referred to obstetrician-led care

P5 115/116: please change psychological problems into past (or previous) or current psychological problems and sexual abuse into a history of sexual abuse

P10, L172 Please finish the sentence … whilst those who were overweight or obese were more likely to be referred.

P12 L185 … and lived in deprivation. You do not know if they actually lived in deprivation, this is an assumption based on their zip/postal code. I suggest this should be changed to …and lived in a deprived area.

P15 L 268-270: During our study period, all women with gestational diabetes were referred to obstetrician-led care, when only a diet was applied to stabilise their glucose values. This sentence seems to have words missing after the 2nd comma? > I think you need to insert the following words: …apart from when only a diet etc etc.

P16 L 274-275: Please be aware of the limitations of your study. Therefor insert “in this particular midwifery practice.” At the end of the sentence.

P16 L286: I think you mean continuity of care instead of Continuous care?

P16 L286-287: is this also the case for women with the described characteristics in your study? Do we know this?

7. PLOS authors have the option to publish the peer review history of their article (what does this mean?). If published, this will include your full peer review and any attached files.

Reviewer #1: No

Reviewer #2: No

---

## [Author Response · Author response to Decision Letter 2]

6 Feb 2023

See also the response letter (Attach Files)

Reviewer 1: 

We checked the subtitles in tables 1a and 1b and corrected these where needed. 

Reviewer 2:

Page 4, last Alinea of introduction: Your focus and your reasons for wanting to carry out this study are now much better explained. However I still have some trouble with the first 2 sentences of this Alinea, they seem to be contradictory. Are lifestyle factors (mentioned in the 2nd sentence) not factors such as BMI and maternal age? And there have been quite few studies so far that have shown an association between stress (psychosocial) and perinatal outcomes. The term lifestyle can be contradicting, therefore we have adjusted the sentences.  There are studies that have shown associations between stress (psychosocial) and perinatal outcomes. Our study does not investigate the association between stress and perinatal outcome, but investigates other psychosocial factors such as psychological problems or sexual abuse and its relation to a referral during the antepartum or intrapartum period. We have clarified this as follows: However, few studies have investigated the association between perinatal outcomes and a wide range of maternal characteristics that are readily available in clinical records, such as a history or presence of psychological problems or sexual abuse [18-25]. Moreover, the associations between specifically non-medical maternal factors and referral towards obstetric care are still unknown. P4, L 86-90

Moreover l87: …and maternal characteristics on non-medical factors > I think it should be: … and non-medical maternal characteristics?  We corrected the sentence based on your comment above: However, few studies have investigated the association between perinatal outcomes and a wide range of maternal characteristics that are readily available in clinical records, such as a history or presence of psychological problems or sexual abuse [18-25]. Moreover, the associations between specifically non-medical maternal factors and referral towards obstetric care are still unknown. P4, L 86-90

P5 L101: I think there are words missing after “and”: Those who had a miscarriage were excluded and were referred to obstetrician-led care in the first trimester etc. This should read: and those who were referred to obstetrician-led care.  We have adjusted this as you suggested: Those who had a miscarriage were excluded and those who were referred to obstetrician-led care in the first trimester or received only postnatal care. P5, L 103-104

P5 115/116: please change psychological problems into past (or previous) or current psychological problems and sexual abuse into a history of sexual abuse.  We have adjusted this as you suggested: Dichotomous variables included: preconception period (≤1 year/>1 year) --the period the woman was trying to conceive--, employment (yes/no), ethnicity (Dutch/non-Dutch) --based on the mothers country of birth --, deprivation (no/yes) -- based on a zip code classified as a deprived area[29]--, smoking (no/yes), psychological problems (past or current) (no/yes), and a history of sexual abuse (no/yes). P5, L 115-122

P10, L172 Please finish the sentence … whilst those who were overweight or obese were more likely to be referred. We have adjusted this as you suggested: Compared to women of healthy weight, those who were underweight were less likely to be referred (0.45; 0.31-0.64), whilst those who were overweight or obese were more likely to be referred (2.29; 1.91-2.74 or 2.65; 2.06-3.42, respectively). P10, L 174-178

P12 L185 … and lived in deprivation. You do not know if they actually lived in deprivation, this is an assumption based on their zip/postal code. I suggest this should be changed to …and lived in a deprived area.  We have adjusted this as you suggested:

Women were also more likely to be referred if their preconception period was longer than one year (1.71; 1.27-2.28), they had a history of psychological problems (1.24; 1.06-1.44), they worked during pregnancy (1.38; 1.19-1.61), and lived in a deprived area (1.23; 1.03-1.46). P12, L 185-189

P15 L 268-270: During our study period, all women with gestational diabetes were referred to obstetrician-led care, when only a diet was applied to stabilise their glucose values. This sentence seems to have words missing after the 2nd comma? > I think you need to insert the following words: …apart from when only a diet etc etc.  Thank you for your suggestion. It is in fact the other way around: women who only needed a diet to stabilise their glucose values were referred to obstetrician-led care during our study period. However, current practice in this region is to not refer them. We have clarified that in the manuscript: During our study period, all women with gestational diabetes were referred to obstetrician-led care. The current policy in this region is that women who only need a diet to stabilise their glucose values are not referred to obstetrician-led care. P16, L 272- 273

P16 L 274-275: Please be aware of the limitations of your study. Therefor insert “in this particular midwifery practice.” At the end of the sentence.  We have adjusted this as you suggested: Our study showed that multiple maternal characteristics are associated with a referral to obstetrician-led care in this particular midwifery practice. P16, L 279-282

P16 L286: I think you mean continuity of care instead of Continuous care?  We have adjusted this as you suggested: Continuity of care might be beneficial for non-medical maternal characteristics, particularly since case-load midwifery care is associated with a lower referral rate [51]. P16, L 291 - 292

P16 L286-287: is this also the case for women with the described characteristics in your study? Do we know this? Yes, women with the described characteristics in our study had access to other caregivers. We have clarified that in the manuscript:

The women in this study had access to other caregivers. P16, L291

---

## [Decision Letter · Decision Letter 3]

27 Feb 2023

Maternal characteristics associated with referral to obstetrician-led care in low-risk pregnant women in the Netherlands: a retrospective cohort study

PONE-D-21-28982R3

Dear Dr. Niessink-Beckers,

We’re pleased to inform you that your manuscript has been judged scientifically suitable for publication and will be formally accepted for publication once it meets all outstanding technical requirements.

Kind regards,

Hanna Landenmark

Staff Editor

PLOS ONE

Additional Editor Comments (optional):

Reviewers' comments:

Reviewer's Responses to Questions

**Comments to the Author**

1. If the authors have adequately addressed your comments raised in a previous round of review and you feel that this manuscript is now acceptable for publication, you may indicate that here to bypass the “Comments to the Author” section, enter your conflict of interest statement in the “Confidential to Editor” section, and submit your "Accept" recommendation.

Reviewer #1: All comments have been addressed

2. Is the manuscript technically sound, and do the data support the conclusions?

Reviewer #1: Yes

3. Has the statistical analysis been performed appropriately and rigorously? 

Reviewer #1: Yes

4. Have the authors made all data underlying the findings in their manuscript fully available?

Reviewer #1: Yes

5. Is the manuscript presented in an intelligible fashion and written in standard English?

Reviewer #1: Yes

6. Review Comments to the Author

Reviewer #1: (No Response)

7. PLOS authors have the option to publish the peer review history of their article (what does this mean?). If published, this will include your full peer review and any attached files.

Reviewer #1: No

---

## [Editor Report · Acceptance letter]

3 Mar 2023

PONE-D-21-28982R3 

Maternal characteristics associated with referral to obstetrician-led care in low-risk pregnant women in the Netherlands: a retrospective cohort study 

Dear Dr. Niessink-Beckers:

I'm pleased to inform you that your manuscript has been deemed suitable for publication in PLOS ONE. Congratulations! Your manuscript is now with our production department. 

Kind regards, 

on behalf of

Dr. Hanna Landenmark 

Staff Editor

PLOS ONE